# TRAINING BINARY NEURAL NETWORKS WITH REAL-TO-BINARY CONVOLUTIONS

**Brais Martinez[1], Jing Yang[1,2,*], Adrian Bulat[1,*] & Georgios Tzimiropoulos[1,2]**
[1] Samsung AI Research Center, Cambridge, UK
[2] Computer Vision Laboratory, The University of Nottingham, UK
`{brais.a,adrian.bulat,georgios.t}@samsung.com`

## ABSTRACT

This paper shows how to train binary networks to within a few percent points ($\sim 3 - 5\%$) of the full precision counterpart. We first show how to build a strong baseline, which already achieves state-of-the-art accuracy, by combining recently proposed advances and carefully adjusting the optimization procedure. Secondly, we show that by attempting to minimize the discrepancy between the output of the binary and the corresponding real-valued convolution, additional significant accuracy gains can be obtained. We materialize this idea in two complementary ways: (1) with a loss function, during training, by matching the spatial attention maps computed at the output of the binary and real-valued convolutions, and (2) in a data-driven manner, by using the real-valued activations, available during inference *prior to* the binarization process, for re-scaling the activations *right after* the binary convolution. Finally, we show that, when putting all of our improvements together, the proposed model beats the current state of the art by more than 5% top-1 accuracy on ImageNet and reduces the gap to its real-valued counterpart to less than 3% and 5% top-1 accuracy on CIFAR-100 and ImageNet respectively when using a ResNet-18 architecture. Code available at `https://github.com/brais-martinez/real2binary`.

## 1 INTRODUCTION

Following the introduction of the BinaryNeuralNet (BNN) algorithm (Courbariaux et al., 2016), binary neural networks emerged as one of the most promising approaches for obtaining highly efficient neural networks that can be deployed on devices with limited computational resources. Binary convolutions are appealing mainly for two reasons: (a) Model compression: if the weights of the network are stored as bits in a 32-bit float, this implies a reduction of $32\times$ in memory usage. (b) Computational speed-up: computationally intensive floating-point multiply and add operations are replaced by efficient `xnor` and `pop-count` operations, which have been shown to provide practical speed-ups of up to $58\times$ on CPU (Rastegari et al., 2016) and, as opposed to general low bit-width operations, are amenable to standard hardware. Despite these appealing properties, binary neural networks have been criticized as binarization typically results in large accuracy drops. Thus, their deployment in practical scenarios is uncommon. For example, on ImageNet classification, there is a $\sim 18\%$ gap in top-1 accuracy between a ResNet-18 and its binary counterpart when binarized with XNOR-Net (Rastegari et al., 2016), which is the method of choice for neural network binarization.

But how far are we from training binary neural networks that are powerful enough to become a viable alternative to real-valued networks? Our first contribution in this work is to take stock of recent advances on binary neural networks and train a very strong baseline which already results in state-of-the-art performance. Our second contribution is a method for bridging most of the remaining gap, which boils down to minimizing the discrepancy between the output of the binary and the corresponding real-valued convolution. This idea is materialized in our work in two complementary ways: **Firstly**, we use an attention matching strategy so that the real-valued network can more

---

* Denotes equal contribution

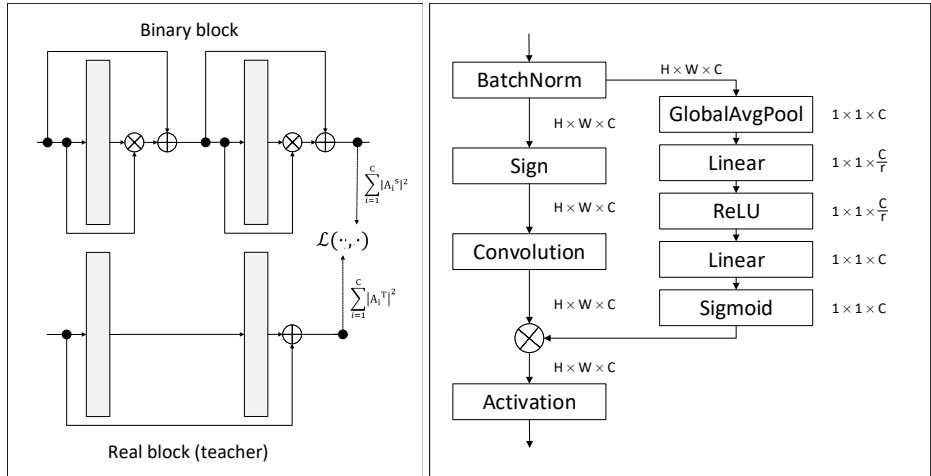

Figure 1: **Left:** The proposed real-to-binary block. The diagram shows how spatial attention maps computed from a teacher real-valued network are matched with the ones computed from the binary network. Supervision is injected at the end of each binary block. See also section 4.2. **Right:** The proposed data-driven channel re-scaling approach. The left-hand side branch corresponds to the standard binary convolution module. The right-hand side branch corresponds to the proposed gating function that computes the channel-scaling factors from the output of the batch normalization. The factor $r$ controls the compression ratio on the gating function, and $H$, $W$ and $C$ indicate the two spatial and the channel dimensions of the activation tensors. See also section 4.3.

closely guide the binary network during optimization. However, we show that due to the architectural discrepancies between the real and the binary networks, a direct application of teacher-student produces sub-optimal performance. Instead, we propose to use a sequence of teacher-student pairs that *progressively* bridges the architectural gap. **Secondly**, we further propose to use the real-valued activations of the binary network, available *prior to* the binarization preceding convolution, to compute scale factors that are used to re-scale the activations *right after* the application of the binary convolution. This is in line with recent works which have shown that re-scaling the binary convolution output can result in large performance gains (Rastegari et al., 2016; Bulat & Tzimiropoulos, 2019). However, unlike prior work, we compute the scaling factors in a data-driven manner based on the real-valued activations of each layer prior to binarization, which results in superior performance.

Overall, we make the following **contributions**:

- We construct a very strong baseline by combining some recent insights on training binary networks and by performing a thorough experimentation to find the most well-suited optimization techniques. We show that this baseline already achieves state-of-the-art accuracy on ImageNet, surpassing all previously published works on binary networks.

- We propose a real-to-binary attention matching: this entails that matching spatial attention maps computed at the output of the binary and real-valued convolutions is particularly suited for training binary neural networks (see Fig. 1 left and section 4.2). We also devise an approach in which the architectural gap between real and binary networks is progressively bridged through a sequence of teacher-student pairs.

- We propose a data-driven channel re-scaling: this entails using the real-valued activations of the binary network *prior* to their binarization to compute the scale factors used to re-scale the activations produced *right after* the application of the binary convolution. See Fig. 1, right, and section 4.3.

- We show that our combined contributions provide, for the first time, competitive results on two standard datasets, achieving 76.2% top-1 performance on CIFAR-100 and 65.4% top-1 performance on ImageNet when using a ResNet-18 –a gap bellow 3% and 5% respectively compared to their full precision counterparts.

## 2   RELATED WORK

While being pre-dated by other works on binary networks (Soudry et al., 2014), the BNN algorithm (Courbariaux et al., 2016) established how to train networks with binary weights within the familiar back-propagation paradigm. The training method relies on a real-valued copy of the network weights which is binarized during the forward pass, but is updated during back-propagation ignoring the binarization step. Unfortunately, BNN resulted in a staggering $\sim 28\%$ gap in top-1 accuracy compared to the full precision ResNet-18 on ImageNet.

It is worth noting that binary networks do have a number of floating point operations. In fact, the output of a binary convolution is not binary (values are integers resulting from the count). Also, in accordance to other low bit-width quantization methodologies, the first convolution (a costly $7 \times 7$ kernel in ResNet), the fully connected layer and the batch normalization layers are all real-valued. In consequence, a line of research has focused on developing methodologies that add a fractional amount of real-valued operations in exchange for significant accuracy gains. For example, the seminal work of XNOR-Net (Rastegari et al., 2016) proposed to add a real-valued scaling factor to each output channel of a binary convolution, a technique that has become standard for binary networks. Similarly, Bi-Real Net (Liu et al., 2018) argued that skip connections are fundamental for binary networks and observed that the flow of full precision activations provided by the skip connections is interrupted by the binary downsample convolutions. This degrades the signal and make subsequent skip connections less effective. To alleviate this, they proposed making the downsample layers real valued, obtaining around $3\%$ accuracy increase in exchange for a small increase in computational complexity.

Improving the optimization algorithm for binary networks has been another fundamental line of research. Examples include the use of smooth approximations of the gradient, the use of PReLU (Bulat et al., 2019), a two-stage training which binarizes the weights first and then the activations (Bulat et al., 2019) and progressive quantization (Gong et al., 2019; Bulat et al., 2019). The work in (Wang et al., 2019) proposed to learn channel correlations through reinforcement learning to better preserve the sign of a convolution output. A set of regularizers are added to the loss term in (Ding et al., 2019) so as to control the range of values of the activations, and guarantee good gradient flow. Other optimization aspects, such the effect of gradient clipping or batch-norm momentum, were empirically tested in (Alizadeh et al., 2019). In section 4.1, we show how to combine many of the insights provided in these works with standard optimization techniques to obtain a very strong baseline that already achieves state-of-the-art accuracy.

While the aforementioned works either maintain the same computational cost, or increase it by a fractional amount, other research has focused instead on relaxing the problem constraints by increasing the number of binary operations by a large amount, typically a factor of 2 to 8 times. Examples include ABC-Net (Lin et al., 2017), the structure approximation of (Zhuang et al., 2019), the circulant CNN of (Liu et al., 2019), and the binary ensemble of (Zhu et al., 2019). Note that the large increase of binary operations diminishes the efficiency claim that justifies the use of binary networks in first place. Furthermore, we will show that there is still a lot of margin in order to bridge the accuracy gap prior to resorting to scaling up the network capacity[1].

The methodology proposed in this paper has some relations with prior work: our use of attention matching as described in section 4.2 is somewhat related to the feature distillation approach of (Zhuang et al., 2018). However, (Zhuang et al., 2018) tries to match whole feature maps of the to-be-quantized network with the quantized feature maps of a real-valued network that is trained in parallel with the to-be-quantized network. Such an approach is shown to improve training of low-bitwidth quantized models but not binary networks. Notably, our approach based on matching attention maps is much simpler and shown to be effective for the case of binary networks.

Our data-driven channel re-scaling approach, described in section 4.3, is related to the channel rescaling approach of XNOR-Net, and also that of (Xu & Cheung, 2019; Bulat & Tzimiropoulos, 2019), which propose to learn the scale factors discriminatively through backpropagation. Contrary to (Xu & Cheung, 2019; Bulat & Tzimiropoulos, 2019), our method is data-driven and avoids using

---

[1]There is also a large body of work focusing on other low-bit quantization strategies, but a review of these techniques goes beyond the scope of this section.

fixed scale factors learnt during training. Contrary to XNOR-Net, our method discriminatively learns how to produce the data-driven scale factors so that they are optimal for the task in hand.

# 3 BACKGROUND

This section reviews the binarization process proposed in (Courbariaux et al., 2016) and its improved version from (Rastegari et al., 2016), which is the method of choice for neural network binarization.

We denote by $\mathcal{W} \in \mathbb{R}^{o \times c \times k \times k}$ and $\mathcal{A} \in \mathbb{R}^{c \times w_{in} \times h_{in}}$ the weights and input features of a CNN layer, where $o$ and $c$ represent the number of output and input channels, $k$ the width and height of the kernel, and $w_{in}$ and $h_{in}$ represent the spatial dimension of the input features $\mathcal{A}$. In (Courbariaux et al., 2016), both weights and activations are binarized using the sign function and then convolution is performed as $\mathcal{A} * \mathcal{W} \approx \text{sign}(\mathcal{A}) \circledast \text{sign}(\mathcal{W})$ where $\circledast$ denotes the binary convolution, which can be implemented using bit-wise operations.

However, this direct binarization approach introduces a high quantization error that leads to low accuracy. To alleviate this, XNOR-Net (Rastegari et al., 2016) proposes to use real-valued scaling factors to re-scale the output of the binary convolution as

$$\mathcal{A} * \mathcal{W} \approx (\text{sign}(\mathcal{A}) \circledast \text{sign}(\mathcal{W})) \odot \mathcal{K}\boldsymbol{\alpha}, \tag{1}$$

where $\odot$ denotes the element-wise multiplication, $\boldsymbol{\alpha}$ and $\mathcal{K}$ are the weight and activation scaling factors, respectively, calculated in Rastegari et al. (2016) in an analytic manner. More recently, Bulat & Tzimiropoulos (2019) proposed to fuse $\boldsymbol{\alpha}$ and $\mathcal{K}$ into a single factor $\Gamma$ that is learned via backpropagation, resulting in further accuracy gains.

# 4 METHOD

This section firstly introduces our strong baseline. Then, we present two ways to improve the approximation of Eq. 1: Firstly, we use a loss based on matching attention maps computed from the binary and a real-valued network (see section 4.2). Secondly, we make the scaling factor a function of the *real-valued* input activations $\mathcal{A}$ (see section 4.3).

## 4.1 BUILDING A STRONG BASELINE

Currently, almost all works on binary networks use XNOR-Net and BNN as baselines. In this section, we show how to construct a strong baseline by incorporating insights and techniques described in recent works as well as standard optimization techniques. We show that our baseline already achieves state-of-the-art accuracy. We believe this is an important contribution towards understanding the true impact of proposed methodologies and towards assessing the true gap with real-valued networks. Following prior work in binary networks, we focus on the ResNet-18 architecture and apply the improvements listed below:

**Block structure**: It is well-known that a modified ResNet block must be used to obtain optimal results for binary networks. We found the widely-used setting where the operations are ordered as BatchNorm $\rightarrow$ Binarization $\rightarrow$ BinaryConv $\rightarrow$ Activation to be the best. The skip connection is the last operation of the block (Rastegari et al., 2016). Note that we use the sign function to binarize the activations. However, the BatchNorm layer includes an affine transformation and this ordering of the blocks allows its bias term act as a learnable binarization threshold.

**Residual learning:** We used double skip connections, as proposed in (Liu et al., 2018).

**Activation:** We used PReLU (He et al., 2015) as it is known to facilitate the training of binary networks (Bulat et al., 2019).

**Scaling factors:** We used discriminatively learnt scaling factors via backpropagation as in (Bulat & Tzimiropoulos, 2019).

**Downsample layers:** We used real-valued downsample layers (Liu et al., 2018). We found the large accuracy boost to be consistent across our experiments (around $3 - 4\%$ top-1 improvement on ImageNet).

We used the following training strategies to train our strong baseline:

**Initialization:** When training binary networks, it is crucial to use a 2-stage optimization strategy (Bulat et al., 2019). In particular, we first train a network using binary activations and real-valued weights, and then use the resulting model as initialization to train a network where both weights and activations are binarized.

**Weight decay:** Setting up weight decay carefully is surprisingly important. We use $1e-5$ when training stage 1 (binary activation and real weights network), and set it to 0 on stage 2 (Bethge et al., 2019). Note that weights at stage 2 are either 1 or $-1$, so applying an $L_2$ regularization term to them does not make sense.

**Data augmentation:** For CIFAR-100 we use the standard random crop, horizontal flip and rotation ($\pm 15°$). For ImageNet, we found that random cropping, flipping and colour jitter augmentation worked best. However, colour jitter is disabled for stage 2.

**Mix-up:** We found that mix-up (Zhang et al., 2017) is crucial for CIFAR-100, while it slightly hurts performance for ImageNet – this is due to the higher risk of overfitting on CIFAR-100.

**Warm-up:** We used warm-up for 5 epochs during stage 1 and no warm-up for stage 2.

**Optimizer:** We used Adam (Kingma & Ba, 2014) with a stepwise scheduler. The learning rate is set to $1e-3$ for stage 1, and $2e-4$ for stage 2. For CIFAR-100, we trained for 350 epochs, with steps at epochs 150, 250 and 320. For ImageNet, we train for 75 epochs, with steps at epochs 40, 60 and 70. Batch sizes are 256 for ImageNet and 128 for CIFAR-100.

## 4.2 REAL-TO-BINARY ATTENTION MATCHING

We make the reasonable assumption that if a binary network is trained so that the output of each binary convolution more closely matches the output of a real convolution in the corresponding layer of a real-valued network, then significant accuracy gains can be obtained. Notably, a similar assumption was made in (Rastegari et al., 2016) where analytic scale factors were calculated so that the error between binary and real convolutions is minimized. Instead, and inspired by the attention transfer method of (Zagoruyko & Komodakis, 2017), we propose to enforce such a constraint via a loss term at the end of each convolutional block by comparing attention maps calculated from the binary and real-valued activations. Such supervisory signals provide the binary network with much-needed extra guidance. It is also well-known that backpropagation for binary networks is not as effective as for real-valued ones. By introducing such loss terms at the end of each block, gradients do not have to traverse the whole network and suffer a degraded signal.

Assuming that attention matching is applied at a set of $\mathcal{J}$ transfer points within the network, the total loss can be expressed as:

$$\mathcal{L}_{att} = \sum_{j=1}^{\mathcal{J}} \|\frac{\mathcal{Q}_S^j}{\|\mathcal{Q}_S^j\|_2} - \frac{\mathcal{Q}_T^j}{\|\mathcal{Q}_T^j\|_2}\|, \qquad (2)$$

where $\mathcal{Q}^j = \sum_{i=1}^c |\mathcal{A}_i|^2$ and $\mathcal{A}_i$ is the $i-$th channel of activation map $\mathcal{A}$. Moreover, at the end of the network, we apply a standard logit matching loss (Hinton et al., 2015).

**Progressive teacher-student:** We observed that teacher and student having as similar architecture as possible is very important in our case. We thus train a sequence of teacher-student pairs that progressively bridges the differences between the real network and the binary network in small increments:
*Step 1*: the teacher is the real-valued network with the standard ResNet architecture. The student is another real-valued network, but with the same architecture as the binary ResNet-18 (e.g. double skip connection, layer ordering, PReLU activations, etc). Furthermore, a soft binarization (a Tanh function) is applied to the activations instead of the binarization (sign) function. In this way the network is still real-valued, but it behaves more closely to a network with binary activations.
*Step 2*: The network resulting from the previous step is used as the teacher. A network with binary activations and real-valued weights is used as the student.
*Step 3:* The network resulting from step 2 is used as the teacher and the network with binary weights and binary activations is the student. In this stage, only logit matching is used.

### 4.3 DATA-DRIVEN CHANNEL RE-SCALING

While the approach of the previous section provides better guidance for the training of binary networks, the representation power of binary convolutions is still limited, hindering its capacity to approximate the real-valued network. Here we describe how to boost the representation capability of a binary neural network and yet incur in only a negligible increment on the number of operations.

Previous works have shown the effectiveness of re-scaling binary convolutions with the goal of better approximating real convolutions. XNOR-Net (Rastegari et al., 2016) proposed to compute these scale factors analytically while (Bulat & Tzimiropoulos, 2019; Xu & Cheung, 2019) proposed to learn them discriminatively in an end-to-end manner, showing additional accuracy gains. For the latter case, during training, the optimization aims to find a set of *fixed* scaling factors that minimize the average expected loss for the training set. We propose instead to go beyond this and obtain discriminatively-trained input-dependent scaling factors – thus, at test time, these scaling factors will *not* be fixed but rather inferred from data.

Let us first recall what the signal flow is when going through a binary block. The activations entering a binary block are actually real-valued. Batch normalization centers the activations, which are then binarized, losing a large amount of information. Binary convolution, re-scaling and PReLU follow. We propose to use the full-precision activation signal, available *prior to* the large information loss incurred by the binarization operation, to predict the scaling factors used to re-scale the output of the binary convolution channel-wise. Specifically, we propose to approximate the real convolution as follows:

$$\mathcal{A} * \mathcal{W} \approx (\text{sign}(\mathcal{A}) \circledast \text{sign}(\mathcal{W})) \odot \boldsymbol{\alpha} \odot G(\mathcal{A}; \mathcal{W}_G), \tag{3}$$

where $\mathcal{W}_G$ are the parameters of the gating function $G$. Such function computes the scale factors used to re-scale the output of the binary convolution, and uses the pre-convolution real-valued activations as input. Fig. 1 shows our implementation of function $G$. The design is inspired by Hu et al. (2018), but we use the gating function to predict ahead rather than as a self-attention mechanism.

An optimal mechanism to modulate the output of the binary convolution clearly should not be the same for all examples as in Bulat & Tzimiropoulos (2019) or Xu & Cheung (2019). Note that in Rastegari et al. (2016) the computation of the scale factors depends on the input activations. However the analytic calculation is sub-optimal with respect to the task at hand. To circumvent the aforementioned problems, our method learns, via backpropagation for the task at hand, to predict the modulating factors using the real-valued input activations. By doing so, more than $1/3$ of the remaining gap with the real-valued network is bridged.

### 4.4 COMPUTATIONAL COST ANALYSIS

Table 1 details the computational cost of the different binary network methodologies. We differentiate between the number of binary and floating point operations, including operations such as skip connections, pooling layers, etc. It shows that our method leaves the number of binary operations constant, and that the number of FLOPs increases by only $1\%$ of the total floating point operation count. This is assuming a factor $r$ of 8, which is the one used in all of our experiments. To put this into perspective, the magnitude is similar to the operation increase incurred by the XNOR-Net with respect to its predecessor, BNN. Similarly, the double skip connections proposed in (Liu et al., 2018) adds again a comparable amount of operations. Note however that in order to fully exploit the computational efficiency of binary convolutions during inference, a specialized engine such as (Zhang et al., 2019; Yang et al., 2017) is required.

## 5 RESULTS

We present two main sets of experiments. We used ImageNet (Russakovsky et al., 2015) as a benchmark to compare our method against other state-of-the-art approaches in Sec. 5.1. ImageNet is the most widely used dataset to report results on binary networks and, at the same time, allows us to show for the first time that binary networks can perform competitively on a large-scale dataset. We further used CIFAR-100 (Krizhevsky & Hinton, 2009) to conduct ablation studies (Sec. 5.2).

| Method | BOPS | FLOPS |
|---|---|---|
| BNN (Courbariaux et al., 2016) | $1.695{\times}10^9$ | $1.314{\times}10^8$ |
| XNOR-Net (Rastegari et al., 2016) | $1.695{\times}10^9$ | $1.333{\times}10^8$ |
| Double Skip ((Liu et al., 2018) | $1.695{\times}10^9$ | $1.351{\times}10^8$ |
| Bi-Real (Liu et al., 2018) | $1.676{\times}10^9$ | $1.544{\times}10^8$ |
| Ours | $1.676{\times}10^9$ | $1.564{\times}10^8$ |
| Full Precision | 0 | $1.826{\times}10^9$ |

Table 1: Breakdown of floating point and binary operations for variants of binary ResNet-18.

## 5.1 COMPARISON WITH THE STATE-OF-THE-ART

Table 2 shows a comparison between our method and relevant state-of-the-art methods, including low-bit quantization methods other than binary.

**Vs. other binary networks:** Our strong baseline already comfortably achieves state-of-the art results, surpassing the previously best-reported result by about $1\%$ (Wang et al., 2019). Our full method further *improves over the state-of-the-art by $5.5\%$ top-1 accuracy*. When comparing to binary models that scale the capacity of the network (second set of results on Tab. 2), only (Zhuang et al., 2019) outperforms our method, surpassing it by 0.9% top-1 accuracy - yet, this is achieved using 4 times the number of binary blocks.

**Vs. real-valued networks:** Our method reduces the performance gap with its real-valued counterpart to $\sim 4\%$ top-1 accuracy, or $\sim 5\%$ if we compare against a real-valued network trained with attention transfer.

**Vs. other low-bit quantization:** Table 2 also shows a comparison to the state-of-the-art for low-bit quantization methods (first set of results). It can be seen that our method surpasses the performance of all methods, except for TTQ (Zhu et al., 2017), which uses 2-bit weights, full-precision activations and 1.5 the channel width at each layer.

## 5.2 ABLATION STUDIES

In order to conduct a more detailed ablation study we provide results on CIFAR-100. We thoroughly optimized a ResNet-18 full precision network to serve as the real-valued baseline.

**Teacher-Student effectiveness:** We trained a real-valued ResNet-18 using ResNet-34 as its teacher, yielding $\sim 1\%$ top-1 accuracy increase. Instead, our progressive teacher-student strategy yields $\sim 5\%$ top-1 accuracy gain, showing that it is a fundamental tool when training binary networks, and that its impact is much larger than for real-valued networks, where the baseline optimization is already healthier.

**Performance gap to real-valued:** We observe that, for CIFAR-100, we close the gap with real-valued networks to about $2\%$ when comparing with the full-precision ResNet-18, and to about $3\%$ when optimized using teacher supervision. The gap is consistent to that on ImageNet in relative terms: $13\%$ and $10\%$ relative degradation on ImageNet and CIFAR-100 respectively.

**Binary vs real downsample:** Our proposed method achieves similar performance increase irrespective of whether binary or real-valued downsample layers are used, the improvement being $5.5\%$ and $6.6\%$ top-1 accuracy gain respectively. It is also interesting to note that the results on the ablation study are consistent for all entries on both cases.

**Scaling factors and attention matching:** It is also noteworthy that the gating module is not effective in the absence of attention matching (see SB+G entries). It seems clear from this result that both are interconnected: the extra supervisory signal is necessary to properly guide the training, while the extra flexibility added through the gating mechanism boosts the capacity of the network to mimic the attention map.

| Method | ImageNet | | |
|---|---|---|---|
| | Bitwidth (W/A) | Top-1 | Top-5 |
| BWN (Rastegari et al., 2016) | 1/32 | 60.8 | 83.0 |
| TTQ (Zhu et al., 2017) | 2/32 | 66.6 | 87.2 |
| HWGQ (Cai et al., 2017) | 1/2 | 59.6 | 82.2 |
| LQ-Net (Zhang et al., 2018) | 1/2 | 62.6 | 84.3 |
| SYQ (Faraone et al., 2018) | 1/2 | 55.4 | 78.6 |
| DOREFA-Net (Zhou et al., 2016) | 2/2 | 62.6 | 84.4 |
| ABC-Net (Lin et al., 2017) | (1/1)×5 | 65.0 | 85.9 |
| Circulant CNN (Liu et al., 2019) | (1/1)×4 | 61.4 | 82.8 |
| Struct Appr (Zhuang et al., 2019) | (1/1)×4 | 64.2 | 85.6 |
| Struct Appr** (Zhuang et al., 2019) | (1/1)×4 | 66.3 | 86.6 |
| Ensemble (Zhu et al., 2019) | (1/1)×6 | 61.0 | – |
| BNN (Courbariaux et al., 2016) | 1/1 | 42.2 | 69.2 |
| XNOR-Net (Rastegari et al., 2016) | 1/1 | 51.2 | 73.2 |
| Trained Bin (Xu & Cheung, 2019) | 1/1 | 54.2 | 77.9 |
| Bi-Real Net (Liu et al., 2018)** | 1/1 | 56.4 | 79.5 |
| CI-Net (Wang et al., 2019) | 1/1 | 56.7 | 80.1 |
| XNOR-Net++ (Bulat & Tzimiropoulos, 2019) | 1/1 | 57.1 | 79.9 |
| CI-Net (Wang et al., 2019)** | 1/1 | 59.9 | 84.2 |
| Strong Baseline (ours)** | 1/1 | 60.9 | 83.0 |
| Real-to-Bin (ours)** | 1/1 | **65.4** | **86.2** |
| Real valued | 32/32 | 69.3 | 89.2 |
| Real valued T-S | 32/32 | 70.7 | 90.0 |

Table 2: Comparison with state-of-the-art methods on ImageNet. ** indicates real-valued down-sample. The second column indicates the number of bits used to represent weights and activations. Methods include low-bit quantization (upper section), and methods multiplying the capacity of the network (second section). For the latter case, the second column includes the multiplicative factor of the network capacity used.

| Method | Stage 1 | Stage 2 |
|---|---|---|
| | Top-1 / Top-5 | Top-1 / Top-5 |
| Strong Baseline | 69.3 / 88.7 | 68.0 / 88.3 |
| SB + Att Trans | 72.2 / 90.3 | 71.1 / 90.1 |
| SB + Att Trans + HKD | 73.1 / 91.2 | 71.9 / 90.9 |
| SB + G | 67.2 / 87.0 | 66.2 / 86.8 |
| SB + Progressive TS | 73.8 / 91.5 | 72.3 / 89.8 |
| Real-to-Bin | **75.0 / 92.2** | **73.5 / 91.6** |
| Strong Baseline** | 72.1 / 89.9 | 69.6 / 89.2 |
| SB + Att Trans** | 74.3 / 91.3 | 72.6 / 91.4 |
| SB + Att Trans + HKD** | 75.4 / 92.2 | 73.9 / 91.2 |
| SB + G** | 72.0 / 89.8 | 70.9 / 89.3 |
| SB + Progressive TS** | 75.7 / 92.1 | 74.6 / 91.8 |
| Real-to-Bin** | **76.5 / 92.8** | **76.2 / 92.7** |
| Full Prec (our impl.) | 78.3 / 93.6 | |
| Full Prec + TS (our impl.) | 79.3 / 94.4 | |

Table 3: Top-1 and Top-5 classification accuracy using ResNet-18 on CIFAR-100. ** indicates real-valued downsample layers. $G$ indicates that the gating function of Sec. 4.3 is used.

## 6 CONCLUSION

In this work we showed how to train binary networks to within a few percent points of their real-valued counterpart, turning binary networks from hopeful research into a compelling alternative to real-valued networks. We did so by training a binary network to not only predict training labels, but also mimic the behaviour of real-valued networks. To this end, we devised a progressive attention matching strategy to drive optimization, and combined it with a gating strategy for scaling the output of binary convolutions, increasing the representation power of the convolutional block. The two strategies combine perfectly to boost the state-of-the-art of binary networks by 5.5 top-1 accuracy on ImageNet, the standard benchmark for binary networks.

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
