# OpenReview forum: "Training binary neural networks with real-to-binary convolutions"
_ICLR.cc/2020/Conference — Accept (Poster)_

### Official Review · AnonReviewer1 · 2019-10-24
**Official Blind Review #1**

**Rating:** 6

**Review:**

This paper greatly reduces the gap between binarized and real valued imagenet, using a variety of techniques. The most significant contributions of this paper are engineering based, and the careful combination and integration of approaches from previous papers. I believe that this is of significant practical importance to the field. I particularly appreciate the effort put into developing a very strong baseline that combined ideas from many previous papers!

My biggest concern is that ResNet is itself a very wasteful architecture in terms of compute and parameter count. If the goal is to develop a compute- and memory-efficient architecture, it would be good to also consider real-valued-network baselines that were proposed with computational and/or memory efficiency as a design goal.

Additionally, the specific choices for the new student-teacher loss, and new scaling network architecture, seem fairly ad-hoc.

Detailed comments:

"this implies a reduction of 32× in memory usage" , assuming the parameter count is held constant.

Fig 1 right: This is motivated in terms of preserving scaling factors that are lost by the binarization, but the functional form for this makes it look a lot like a learned gating operation. If the sigmoid is dropped from the architecture, does performance worsen? It would be nice to see some discussion of the degree to which this is helpful because it reverses information loss due to binarization, vs. introduces a new architectural feature which is itself helpful.

Add a sentence describing what "double skip connections" are. I wasn't familiar with this phrase.

eq. 2:
This functional form is pretty weird.
Why is Q a square norm rather than a norm? Square error on an already-squared property is an unusual choice.
Why is the denominator itself a norm? Taking the norm of a square norm is similarly an unusual choice. (eg, why not just take an average or sum over Q)
Say what Q_s and Q_t are (student and teacher network from context)

"thus, at test time, these scaling factors will not be fixed but rather inferred from data" nit: Would not generally call this an inference process. "Inference" typically refers to values that are computed indirectly (eg by Bayesian reasoning), while in this case the values are computed directly. Would rather say that scaling factors are a function of data, or are determined by data, or similar.

"By doing so, more than 1/3 of the remaining gap with the real-valued network is bridged." text is shifting back and forth between using % and fractional gap to describe benefits. Would just use one measure consistently.

Computational cost analysis:
This is very useful.
Note though that ResNet is a very wasteful architecture in terms of compute! It would be good to include a comparison to imagenet architectures that have computational and memory efficiency as a design goal. (eg, MobileNet comes to mind)

Very nice on the ablation studies.

**Experience Assessment:**

I have read many papers in this area.

**Review Assessment: Checking Correctness Of Derivations And Theory:**

N/A

**Review Assessment: Checking Correctness Of Experiments:**

I assessed the sensibility of the experiments.

**Review Assessment: Thoroughness In Paper Reading:**

I read the paper at least twice and used my best judgement in assessing the paper.

---

> ### Author Response · Authors · 2019-11-15
> **Response to Reviewer #1**
>
>
> R1.1 “My biggest concern is that ResNet is itself a very wasteful architecture in terms of compute and parameter count. If the goal is to develop a compute- and memory-efficient architecture, it would be good to also consider real-valued-network baselines that were proposed with computational and/or memory efficiency as a design goal.”
>
> Training a highly accurate binary ResNet has been an open problem for many years and to our knowledge, our work is the first one that bridges most of the accuracy gap. We believe we are taking a very important step forward considering the actual current state of research on the topic. Moving into more compute- and memory-efficient architectures is definitely one of our next goals. To our knowledge we are not aware of any existing work aiming at solving this very challenging problem.
>
>
> R1.2 On  specific choices for student-teacher, and new scaling network, being adhoc
>
> We believe that the loss of Eq. (2) is relatively straightforward: its objective is to transfer normalized attention maps from the teacher to the student. See  also R4.2.
>
> Furthermore, please see R3.7 for a more detailed discussion on 2-stage vs. multi-stage teacher-student optimization strategy. Moreover, the architecture for the scaling block works as follows: it computes a global spatial average (since we’re interested in computing a single factor per channel) which is then processed by 2 FC layers implementing a simple bottleneck. Thank you for this comment we will further clarify in the paper.
>
>
> R1.3 On "this implies a reduction of 32× in memory usage", assuming the parameter count is held constant:
>
> We hold the parameter count constant, except for the scaling function parameters. which account to around 200k extra parameters, less than 2% those of a a ResNet18, which has 11.7M.
>
>
> R1.4: On “Fig 1 right: This is motivated in terms of preserving scaling factors that are lost by the binarization, but the functional form for this makes it look a lot like a learned gating operation. If the sigmoid is dropped from the architecture, does performance worsen? It would be nice to see some discussion of the degree to which this is helpful because it reverses information loss due to binarization, vs. introduces a new architectural feature which is itself helpful.”
>
> This is an interesting suggestion, and we thank the reviewer for taking the time to think about the methodology. We have tried without the sigmoid at the end and found performance to degrade by a large margin for the final model (stage 2), while stage 1 works only slightly worse (0.5% worse).
>
>
> R1.5 On what "double skip connections" are.
>
> Apologies, we will add the following to the manuscript to clarify:
>
> “A double skip connection modifies the ResNet basic block to have a skip connection per convolution rather than one per block. This is illustrated in Fig.1, left.”
>
>
> R1.6 On the functional form of Eq. 2:
>
> Q^j(h,w) captures the energy of the activations at location h,w. The important thing to note here is that Q^j is normalized in Eq. 2 by its norm. This is required so that the scaling of the activations does not change the representation. Given that they are normalized, the L2 norm is a way of comparing them - irrespective of how they were computed.
>
>
> R1.7: On using the term "Inference":
>
> Changed to “are determined by data”, thank you!
>
>
> R1.8: On using the fractional vs percentages.
>
> We agree that, although they give complementary information, changing between the two is not good practice. We’ve given the paper a pass to homogenize this.
>
>
> R1.9: On comparing with computational and memory efficiency architectures
>
> In our ResNet-18 implementation, we used a very expensive 7x7 convolution for the first layer (118M FLOPs). Note here that the first convolution is always real-valued even for binary networks. This was done in order to make a fair comparison with previous work on binary networks,  which also use the same ResNet-18 implementation. However, the first convolution can be replaced with negligible drop in performance by more efficient layers (e.g. like the ones in the daBNN paper), which reduces the complexity of the first layer to 42M FLOPs. Considering this, the comparison with MobileNet-V2, one of the state-of-the-art efficient architectures is:
>
> MobileNet-V2:                                             --                300M FLOPs
> Ours                                                   1676M BOPs      150M FLOPs
> Ours with Stem as in daBNN:        1676M BOPs        80M FLOPs
>
>
> We note that on a modern x64 CPU, using bit-packing and SIMD the theoretical throughput is around 64 binary instructions per clock, making a FLOP roughly equivalent with 64BOPS. This is in line with the theoretical speedups report in XNOR-Net (Rastegari et al). Finally, we note that the model we trained for our paper produced accuracy of 65.4% while a fully real-valued MobileNet-v2 gets 72.0%.

---

### Official Review · AnonReviewer2 · 2019-10-29
**Official Blind Review #2**

**Rating:** 6

**Review:**

This paper is on building binary network. The steps for building binary network takes several components: traditional strategy to binary/optimize a model (like data augmentation,  binary initialization using 2-stage optimization, etc), real-to-binary attention matching that tries to match the output of real values and binarized model, and data-driven channel rescaling to better approximate real convolutions. All these components together makes a strong binary network.

Although there are so many steps/tricks mentioned in the paper, I think the explanation and reason for each step is easy to understand. The outcome of the model is quite impressive-- 5% improvement over the best binary model.

It would be interesting to compare with some other compression techniques, like low-rank, sparsity, weight sharing, etc. Or it will be also interesting to see how these techniques can combine with binary model to further compress the model.

Also it would be interesting to see how the latency changes using the proposed binary model. As I can see from Table 1, which outlines the FLOPS and BOPS, to my understanding BOPS is much faster than FLOPS, so in the latency wise, the proposed model will be much faster than the original model for inference. Therefore I am looking forward to the real-timing results.

**Experience Assessment:**

I have published in this field for several years.

**Review Assessment: Checking Correctness Of Derivations And Theory:**

I assessed the sensibility of the derivations and theory.

**Review Assessment: Checking Correctness Of Experiments:**

I assessed the sensibility of the experiments.

**Review Assessment: Thoroughness In Paper Reading:**

I read the paper at least twice and used my best judgement in assessing the paper.

---

> ### Public Comment · ~da_quexian1 · 2019-11-07
> **About latency on real devices**
>
> I'm the author of daBNN [1], a highly optimized BNNs inference framework for mobile.
>
> It is not so easy to measure the latency of BNNs on real devices. The latency heavily depends on the implementation. One needs to re-implement convolutions using bit-wise operations by assembly if he/she wants them to be fast. For example, on BMXNet [2], which only uses pure c/c++, BNNs are even slower than full-precision networks in most cases, despite their high theoretical speedup.
>
> As a result, it is normal that the authors of this paper didn't report the real latency -- there is no highly optimized implementation for BNNs until daBNN.
>
> However, I'd like to recommend daBNN if the authors are also curious about the real latency of their BNNs. daBNN is many times faster than BMXNet and full-precision TF Lite on mobile phones. I will provide the necessary help on GitHub if someone wants to use daBNN and has trouble.
>
> [1] Jianhao Zhang, et.al. daBNN: A Super Fast Inference Framework for Binary Neural Networks on ARM devices, 2019.
> [2] Haojin Yang, et.al. BMXNet: An Open-Source Binary Neural Network Implementation Based on MXNet, 2017

---

> ### Author Response · Authors · 2019-11-15
> **Response to Reviewer #2**
>
> R2.1: On comparing with other compression techniques, and exploring their combination with binarization:
>
> In general, the 32x compression provided by binary models is hard to beat. However, we opted not to include comparison with non-binary methods because we felt that providing a fair comparison between different approaches would be extremely complex. The suggestion for a combination is very interesting but it requires a comprehensive investigation which could fit into a new paper.
>
>
> R2.2: On latency and real-timing results:
>
> Please see R4.4: practical performance depends dramatically on hardware implementation which  is not available to us at the moment.

---

### Official Review · AnonReviewer3 · 2019-10-30
**Official Blind Review #3**

**Rating:** 6

**Review:**

This paper studies the problem of training binary neural networks. The authors first provide a strong baseline by assembling a group of training techniques that appeared in recent work that achieves state-of-the-art performance. Then the authors proposed two methods to further boost the performance gain. The first method is to use a teacher-student mechanism that uses a fully real-valued network to teach a binary network. The process is divided into three stages involving two intermediate models to reduce the gap within each teacher-student pair. The second method is to learn a re-scale factor for binary activations using real-valued activations from the previous block. Experiments show that the proposed methods improves the performance on ImageNet and CIFAR-100.
The experimental results seem promising. The proposed model reduces the gap to real-valued network to within 3-5%. However, the novelty of the paper is limited and why the proposed methods would help increase the performance gain is not well demonstrated. The teacher-student model is a well-known technique for vision tasks. The authors observed in Section 4.2 that it is very important for the teacher and student to have similar architectures, but did not explain the more important question that why a real-valued network would be able to teach a binary network, since they have quite different information flow. For re-scaling, the authors did not give a detailed comparison between their approach and previous work, and it is not clear how the data-driven way helps. As the ablation study shows the gating function actually hurts for binary down-sampling layers.
The writing of the paper needs improvement. A workflow/framework/algorithm description is helpful to better understand the whole framework, and the methodology part in Section 4 requires more details. Some notations need to be defined or clarified. For instance, in Figure 1 Left, what is A? The definition is given only in Section 4.2, where it is not stated in detail either. In Figure 1 Right, what is r? In Table 3, what do the abbreviations mean respectively?
Some specific questions:
- Why the real-valued teacher can help train the binary network while they have different information flow? What is the intuition behind the consistency assumption?
- The authors did not visually show the maps of real-valued and binary activations. How are they aligned in the proposed framework? And are they more similar with each other compared with previous approaches?
- In Section 4.1 for Initialization a 2-stage optimization strategy is used, while in Section 4.2 a multi-stage teacher-student optimization strategy is used. How are the two strategies combined?


**Experience Assessment:**

I do not know much about this area.

**Review Assessment: Checking Correctness Of Derivations And Theory:**

N/A

**Review Assessment: Checking Correctness Of Experiments:**

I assessed the sensibility of the experiments.

**Review Assessment: Thoroughness In Paper Reading:**

I read the paper at least twice and used my best judgement in assessing the paper.

---

> ### Author Response · Authors · 2019-11-15
> **Response to Reviewer #3 (part 1/2)**
>
>
> R3.1: On paper novelty, and teacher-student
>
> To our knowledge, we are 1) the first to construct and train appropriately a very strong baseline which outperforms all previous work, and then 2) propose the ideas of Sections 4.2 and 4.3 that show how to surpass this baseline by a large margin (~4.5% on ImageNet, ~5.5% on CIFAR-100).
>
> As pointed out by the reviewer there are several papers on student-teacher networks, but to our knowledge none of them has been successfully applied for training a binary network reporting such high accuracy gains as in our work. To this end, we have devised the multi-stage training described in Section 4.2, which results in much improved performance.
>
> Moreover, in Section 4.3 we propose a novel architecture (the input-dependent scaling function) that improves performance by a large margin for both CIFAR-100 and ImageNet, on top of an already well-optimized network (2.7% and 2.3% extra top-1 performance on ImageNet for binary and real-valued downsample respectively).
>
>
> R3.2: On why a real-valued network would be able to teach a binary network, since they have quite different information flow:
>
> Thank you for providing this comment. It is not true that the binary and real networks have such different flow.  When analyzed, the features from a binary network follow the same typical expected structure: the low level features represent edges while the higher ones more abstract, class specific, concepts. As such it is not surprising that by having an explicit reference to the desired distribution of a real-valued model we can further boost the performance.
>
> That said, we agree with the reviewer that there is a gap between the two that renders  a direct application of standard teacher-student difficult. That is exactly why we devised the multi-stage optimization. Through this procedure, the teacher network used to train the stage 1 network (binary activations, real-valued weights) will have a much closer flow and thus provide better improvement. Similarly, for stage 2, we use stage 1 network as teacher as this has a closer flow than a full precision network.
>
> We will show feature maps from the real and binary network in the supplementary material, also providing discussion based on your question.
>
>
> R3.3: On re-scaling, and comparison of our approach with previous work:
>
> Recent papers have shown that learning scaling values through backpropagation improves performance (see Xu & Cheung 2019 and Bulat & Tzimiropoulos 2019 on our submission). Our baseline does indeed use such scaling variables. When we show improvements due to the input-dependent scaling function, we are comparing against the best-performing scaling strategy to date.
>
> The following paragraph from our paper describes the differences between the scaling factors used in prior work and in our paper:
>
> “Previous works have shown the effectiveness of re-scaling binary convolutions with the goal of better approximating real convolutions and in turn achieving large accuracy gains. XNOR-Net (Rastegari et al., 2016) proposed to compute these scale factors analytically while (Bulat & Tzimiropoulos,2019; Xu & Cheung, 2019) proposed to learn them discriminatively in an end-to-end manner, showing additional accuracy gains. For the latter case, during training, the optimization aims to find a set of fixed scaling factors that minimize the average expected loss for the training set. We propose instead to go beyond this and obtain input-dependent scaling factors – thus, at test time, these scaling factors will not be fixed but rather inferred from data”
>
> Finally, the following paragraph from our paper provides intuition about why our method is needed and works:
>
> “Let us first recall what the signal flow is when going through a binary block. The activations entering a binary block are actually real-valued. Batch normalization centers the activations, which are then binarized, thus losing a large amount of information. Binary convolution, re-scaling and eventually PReLU follow. We propose to use the full-precision activation signal, prior to the large information loss incurred by the binarization operation, to predict the scaling factors used to re-scale the output of the binary convolution channel-wise.”
>
> “Why is this important?: An optimal mechanism to modulate the output of the binary convolution clearly should not be the same for all examples as in Bulat & Tzimiropoulos (2019) or Xu & Cheung(2019). Note that in Rastegari et al. (2016) the computation of the scale factors depends on the input activations. However the analytic calculation is sub-optimal with respect to the task at hand. To circumvent the aforementioned problems, our method learns, via backpropagation and for the task at hand, to predict the modulating factors using the real-valued input activations. By doing so, more than 1/3rd of the remaining gap with the real-valued network is bridged.”

---

> > ### Author Response · Authors · 2019-11-15
> > **Response to Reviewer #3 (part 2/2)**
> >
> >
> > R3.4: On not being clear how the data-driven way helps:
> >
> > The scale function is the only difference between the full Real-to-Bin and the strong baseline + att trans. + kd entries on Table 3. Below are the results for ImageNet (the binary downsample results were not included in the original submission):
> >
> > ImageNet - Binary downsample                               top1/top5
> > Strong baseline (SB)                                                     57.746 / 80.406
> > SB + Att trans + KD (TS)                                                 59.386 / 81.854
> > Real-to-Bin (SB + TS + Scale Gating function)           62.106 /  83.996
> >
> > ImageNet - Real downsample                                    top1/top5
> > Strong Baseline (SB)                                                     60.9 / 83.0
> > SB + Att trans + KD (TS)                                                 63.1 / 84.8
> > Real-to-Bin (SB + TS + Scale Gating function)           65.4 / 86.2
> >
> > This shows a 2.7% gain for the binary downsample gain in terms of top-1 accuracy, and a 2.3% for the real downsample case. This is a very large performance gain, especially considering that the baseline is already extremely well optimized.
> >
> > Similarly, for CIFAR-100:
> >
> > CIFAR - Binary downsample                                       top1/top5
> > Strong baseline (SB)                                                    68.03 / 88.28
> > SB + Att trans + KD (TS)                                                71.93 / 90.93
> > Real-to-Bin (SB + TS + Scale Gating function)          73.49 / 91.55
> >
> >
> > CIFAR - Real downsample                                           top1/top5
> > Strong baseline (SB)                                                    70.49 / 88.81
> > SB + Att trans + KD (TS)                                                74.62 / 91.79
> > Real-to-Bin (SB + TS + Scale Gating function)          76.15 / 92.67
> >
> > We found however that optimization of the scaling function is hard without the extra help from the teacher student. That is shown in Table 3 as SB + G (strong baseline + Gating function). We believe that we were not clear enough there, as this same comment was raised in a previous post by Zhaohui Yang. We apologize for the confusion.
> >
> >
> > R3.5: On improving the writing of the paper:
> >
> > We are giving the paper a thorough pass and will improve the clarity of the text.
> >
> >
> > R3.6: On real-valued and binary maps, and their visualizations:
> >
> > We thank the reviewer for the suggestion regarding their visualization, we will include a figure in the supplementary material.
> >
> > Yes, maps are more similar. The difference with real-valued attention maps for the validation set goes down from 0.0073 to 0.0012 - more than 6 times lower.
> >
> > We compare the activation maps at the end of each stage. Thus, they both have the same shapes and encode information at corresponding points of the network.
> >
> > Please see also R3.2.
> >
> >
> > R3.7:  On 2-stage vs. multi-stage teacher-student optimization strategy, and how they are combined:
> >
> > We will clarify this, as we agree that this could be explained better in the text:
> > The idea of the 2-stage optimization strategy is to first train a model with binary activations/real-valued weights, and then train a model with both activations and weights binarized using the stage 1 model as initialization.  When using teacher-student, we do train a sequence of real-valued teacher-student networks so that the teacher used for stage 1 is more adequate.

---

### Official Review · AnonReviewer4 · 2019-10-30
**Official Blind Review #4**

**Rating:** 6

**Review:**

A. Summary

Problem:
Binary NNs promise to make neural networks compatible with devices that only have access to limited computational resources (fundamental to embed NNs in mobiles or IoT devices). However, the loss of computational accuracy comes with a great loss of performances. Designing the right learning algorithm and the right binary architecture remains an open issue.

Contributions:
1. this paper reviews the current literature in Binary Neural Networks and the authors compile the existing methods to build a strong baseline. The strong baseline outperforms existing methods, which is impressive in itself.
2. the authors introduce a novel layer-wise objective that pushes the binary activations to match the real activations, which are given by a teacher model (real-to-binary). This is simpler and more efficient than existing alternatives.
3. they propose to train the real-to-binary model using a multi-stage teacher-student procedure.
4. the authors introduce a data-dependent re-scaling term for the binary activations.

Experiments:
1. SOTA on ImageNet for BinaryNNs: The combined methods allow bridging the gap between real-valued and binary-valued classifiers on ImageNet (65.4% vs 69.3% top-1 acc).
2. Comparisons with existing methods: On ImageNet, the model is compared with a complete list of alternative methods (low-bit quantization, larger binary nets, binary nets and real-valued nets). This is however unclear how the method compares with TTQ.
3. An ablation study is performed on CIFAR-100. It tests the gains that come with the attention matching, the data-dependent gating mechanism and the multi-stage teacher-student mechanism.

B. Decision
6: Weak Accept.

C. Argumentation
The paper clearly states the problem and what are the contributions. The solution is mostly iterative but clearly brings the binary NNs one step forward. The claims are supported by a comparison with a great variety of baselines and an ablation study. Furthermore, it is laudable that great efforts were put into designing such a strong baseline.

However, the paper could be easier to read and some points remain unclear:
1. Is it possible to compare TTQ on the same scale? this is difficult to precisely asses how real-to-binary convolutions compete with this method in the paper.
2. The equation 2. is intuitive yet not perfectly clear. What are the "transfer points"? Why using such a normalization?


D. Feedback
1. Data augmentation and Mix-up: is it necessary to use them here as they should yield improvements for all methods? `
2. table 1: is it possible to include TTQ? suggestion: is it possible to add a 4th column that measures the overall performance (estimated runtime, speedup?)
3. table 3: please define all the abbreviations.

E. Question
1. Could you please draw a more precise comparison between TTQ and your method?

**Experience Assessment:**

I do not know much about this area.

**Review Assessment: Checking Correctness Of Derivations And Theory:**

I assessed the sensibility of the derivations and theory.

**Review Assessment: Checking Correctness Of Experiments:**

I assessed the sensibility of the experiments.

**Review Assessment: Thoroughness In Paper Reading:**

I read the paper at least twice and used my best judgement in assessing the paper.

---

> ### Author Response · Authors · 2019-11-15
> **Response to Reviewer #4**
>
> R4.1: On comparison with TTQ:
>
> It is not straightforward to compare TTQ with our method because of the following reasons:
> 1) Our setting is significantly more challenging: TTQ uses 2 bits for the weights and full precision for the activations. We use 1-bit for both weights and activations.
> 2) TTQ uses a bigger network, called ResNet-18B, for their ImageNet experiments, which has 50% more channels than the standard ResNet18 used in our work.
>
> In any case, the main methodological novelty of TTQ, learning the quantization points, is also standard on binary networks since the XNorNet paper (Rasteragi et al, ECCV 2016).
>
>
> R4.2: On equation 2, "transfer points", and normalization:
>
> A transfer point is the location within the network at  which the attention maps are computed, and compared between the teacher and the student. We used 4 transfer points each at the end of a stage (spatial resolution). Normalization is used so that the attention maps do not depend on the magnitude of the response maps. These magnitudes are different throughout the network. Thank you for this comment, we will clarify this in the paper.
>
>
> R4.3: On Data augmentation and Mix-up:
>
> We believe it is necessary to use them to show how far binary networks can get in terms of accuracy. If previous methods didn’t use them, we believe it is something worth reporting in our paper. Furthermore, all data augmentations used are standard for the full precision case.
>
> Moreover, we used these techniques to train our strong baseline upon which we improve later on by 4-5%. So overall, we believe comparison is fair. Exactly the same mixup and data augmentation techniques are considered for binary and real-valued networks.
>
>
> R4.4: On including TTQ + adding a 4th column that measures the overall performance:
>
> TTQ has no BOPS (xnor + popcount ), as TTQ has real-valued activations, and their flop count would be the same as for the real-valued base architecture used. Please see also R4.1.
>
> Unfortunately we cannot provide speed-up related figures because we don’t have a hardware implementation. This is crucial to achieving the speed-ups, which depend on the implementation. Existing deep learning frameworks don’t support this kind of quantization (they support up to 8 bits).
>
>
> R4.5: On table 3 and abbreviations:
> We are currently revising the manuscript, thank you for pointing this out.
>
> R4.6: On comparison with TTQ:
> Please see R4.1 and R4.4.

---

> > ### Comment · AnonReviewer4 · 2019-11-15
> > **All my questions have been answered.**
> >
> > R4.1: yes, 2bits NNs and 1bit NNs are not straightforwardly comparable. R4.4 this makes the comparison between models difficult as the overall performance (e.g. latency) highly depends on the implementation (see the answer from Da Quexian: about latency on real devices). R4.3 as data augmentation and mix-up are common practice, you are indeed bound to use them.
> >
> > Based on the above answer and the overall discussion, it seems that the authors should be able to resolve most of the problems mentioned by the reviewers. Furthermore, the feedback should allow improving the writing, which remains one important downside of the paper.

---

### Public Comment · ~Zhaohui_Yang1 · 2019-10-31
**Interesting paper**

Hi,

I found this paper to be very interesting, and the results are very strong. This paper proposed a strong baseline, a multi-stage distillation strategy, and a re-scale branch on the shortcut. Some concerns about the paper and experiments are as follow,

About baseline:
1. The ablation study of the results on the ImageNet using ResNet18?
1.1 Strong Baseline - 2 stage optimization strategy
1.2 Strong Baseline - 2 stage optimization strategy and replace bn-conv-prelu by conv-bn
2. What is the Binarization function? The sign function?

About Re-Scale branch:
1. From Tab. 3, it seems that the re-scale does not guarantee improvements on accuracy. What is the result of ImageNet w/o the re-scale factor?
2. What is the hyper-parameter $r$ used in the re-scale branch, and what about the number of parameters increase?
3. The re-scale branch used for the downsampling layers' shortcut or all the shortcuts?

Thanks!

---

> ### Author Response · Authors · 2019-11-01
> **Thank you for the comments**
>
> Hi Zhaohui,
>
> thank you for taking the time to read our paper, for your positive words, and for the good questions. Please do let us know if some point is not sufficiently clarified.
>
> Q1: Ablation on ImageNet
>
> R1: The ablation study is done on CIFAR100 for computational reasons. We test two settings (binary downsample and real downsample) and each model involves (at least) two training rounds. That said, we have some results on ImageNet on what we believe are the most relevant configurations of the ablation study. They however are not as comprehensive as for CIFAR100 due to the aforementioned reasons:
>
> Binary downsample (top1/top5)
> Strong baseline                                  57.746 / 80.406
> Strong baseline + Att trans + KD     59.386 / 81.854
> Real-to-Bin (full model)                    62.106 /  83.996
>
> Real downsample (top1/top5)
> Strong Baseline                                  60.9 / 83.0
> Strong baseline + Att trans + KD     63.1 / 84.8
> Real-to-Bin                                          65.4 / 86.2
>
> The difference between the full model (referred to as "Real-to-Bin") and the "Strong baseline + Att trans + KD" is the use of the gating function to re-scale the convolution output. This gives approx 2.7 and 2.3 extra top1 for the binary/real downsample cases.
> Also, it is possible to see that the binary downsample follows a similar trend to the real downsample, with the baseline also being already SOTA and each configuration improving results by a large margin
>
>
> Q2: Binarization function used
>
> R2: We simply use the sign function. We are aware that several works have reported improvements when using approximations, and that might help our method even further. However, we haven't experimented with these options.
>
>
> Q3: result of ImageNet w/o the re-scale factor
>
> R3: the results included above (on R1) clarify the crucial importance of the re-scale branch. Specifically:
>
> binary downsample without/with:
>     59.4 vs 62.1 (2.7 top1 improvement)
>
> real downsample without/with:
>     63.1/65.4 (2.3 top1 improvement)
>
> It is true that table 3 might give the impression that the gating function does not work on some cases due to the "SB+G" entry. We should have made this clearer on the text. What we wanted to show with the "SB+G" entry is that attention transfer is fundamental so that the re-scale branch has healthy gradients and a clear target. Otherwise, the optimization is not successful.
>
> Table 3 shows that the re-scale branch adds 1.5% on top of the best-performing CIFAR-100 model (74.62 without, 76.15 with). This is consistent roughly with the performance on imagenet - it is just that the gap with respect to the full precision network on CIFAR is smaller
>
>
> Q4: hyper-parameter $r$ and parameters increase
>
> R4: The parameter increase is shown in table 1, where operations are split between full precision operations and binary operations. The difference when using the gating function vs. not using it is 1.544*10^8 vs 1.564*10^8 FLOPs (binary operations stay the same at 1.676*10^9). Thus, it is a very marginal increase on computational cost in exchange for >2 top1 performance increase
>
> The reduction ratio is set to 8
>
>
> Q5: The re-scale branch used for the downsampling layers' shortcut or all the shortcuts?
>
> R5: Downsampling layers do not have the gating (re-scaling) layer. It is used on all of the 3x3 convolutions and only on those.

---

> > ### Public Comment · ~Zhaohui_Yang1 · 2019-11-01
> > **I'm clear now, thanks!**
> >
> > I'm clear now, thanks!

---

### Decision · Program_Chairs · 2019-12-19

**Decision:**

Accept (Poster)

**Comment:**

This paper proposes methodology to train binary neural networks.

The reviewers and authors engaged in a constructive discussion. All the reviewers like the contributions of the paper.

Acceptance is therefore recommended.